# Hormetic Effect of Glyphosate on the Morphology, Physiology and Metabolism of Coffee Plants

**DOI:** 10.3390/plants12122249

**Published:** 2023-06-08

**Authors:** Renato Nunes Costa, Natalia da Cunha Bevilaqua, Fábio Henrique Krenchinski, Bruno Flaibam Giovanelli, Vinicius Gabriel Caneppele Pereira, Edivaldo Domingues Velini, Caio Antonio Carbonari

**Affiliations:** Department of Crop Science, College of Agricultural Sciences, São Paulo State University (Universidade “Júlio de Mesquita Filho” UNESP), Botucatu 18610-034, SP, Brazil; nataliacunha_8@hotmail.com (N.d.C.B.); fhkrenchinski@gmail.com (F.H.K.); bfgiovanelli@yahoo.com.br (B.F.G.); viniciuscanepp@gmail.com (V.G.C.P.); edivaldo.velini@unesp.br (E.D.V.); caio.carbonari@unesp.br (C.A.C.)

**Keywords:** hormesis, underdoses, shikimic acid pathway, growth stimulation, photosynthesis

## Abstract

Glyphosate is a nonselective herbicide of systemic action that inhibits the enzyme 5-enolpyruvylshikimate-3-phosphate synthase, thus compromising amino acid production and consequently the growth and development of susceptible plants. The objective of this study was to evaluate the hormetic effect of glyphosate on the morphology, physiology, and biochemistry of coffee plants. Coffee seedlings (*Coffea arabica* cv Catuaí Vermelho IAC-144) were transplanted into pots filled with a mixture of soil and substrate and subjected to ten doses of glyphosate: 0, 11.25, 22.5, 45, 90, 180, 360, 720, 1440, and 2880 g acid equivalent (ae) ha^−1^. Evaluations were performed using the morphological, physiological, and biochemical variables. Data analysis for the confirmation of hormesis occurred with the application of mathematical models. The hormetic effect of glyphosate on coffee plant morphology was determined by the variables plant height, number of leaves, leaf area, and leaf, stem, and total dry mass. Doses from 14.5 to 30 g ae ha^−1^ caused the highest stimulation. In the physiological analyses, the highest stimulation was observed upon CO_2_ assimilation, transpiration, stomatal conductance, carboxylation efficiency, intrinsic water use efficiency, electron transport rate, and photochemical efficiency of photosystem II at doses ranging from 4.4 to 55 g ae ha^−1^. The biochemical analyses revealed significant increases in the concentrations of quinic acid, salicylic acid, caffeic acid, and coumaric acid, with maximum stimulation at doses between 3 and 140 g ae ha^−1^. Thus, the application of low doses of glyphosate has positive effects on the morphology, physiology, and biochemistry of coffee plants.

## 1. Introduction

In plants, the stimulatory effect caused by low doses of herbicides is [1,2,3,4,5,6,7,8] called hormesis and is characterized by a biphasic effect on a dose–response curve, i.e., the same herbicide has a beneficial effect on plants at low doses but an inhibitory effect at high doses [1,2,3,4,5,6,7,8]. Although stimulating plant growth is not the purpose of applying herbicides, these plants are likely to come into contact with doses that can cause the hormetic effect [9,10]. These doses can reach target and nontarget plants through herbicide drift, irregular application, product dilution, absorption of low doses by the soil, foliar contact between treated and untreated plants, the interception of spray solution by higher plants, and purposeful application.

Glyphosate is a nonselective systemic herbicide that inhibits the enzyme 5-enolpyruvylshikimate-3-phosphate synthase (EPSPS). EPSPS acts directly on the shikimate pathway, thus inhibiting the synthesis of the essential aromatic amino acids phenylalanine, tyrosine, and tryptophan, which are precursors of other compounds, such as lignin, alkaloids, flavonoids, and benzoic acids, that are indispensable for protein synthesis and plant growth [11,12]. Glyphosate may directly affect plant photosynthesis by reducing the activity of the enzyme ribulose-1,5-bisphosphate carboxylase/oxygenase (Rubisco) and the 3-phosphoglyceric acid, decreasing chlorophyll synthesis, and interfering with the organization of the photosynthetic apparatus [13,14].

One of the most-studied herbicides that exhibits this behavior is glyphosate [1,5,8,15,16]. Approaches range from the possible influences on the evolution of herbicide resistance [8,17] to the attenuation of environmental stresses on plants [7,18]. In a literature review conducted by Brito et al. [1], the authors concluded that glyphosate application to susceptible plants at low doses can stimulate plant growth and increase the rate of electron transport, CO_2_ assimilation, stomatal conductance, and transpiration rate, in addition to inducing acid accumulation, increasing and accelerating flowering, modifying gene expression, increasing crop yield, and improving biomass quality. However, these same authors reported that most studies explain this phenomenon through morphological variables.

The hormetic effect of glyphosate is observed at different dose ranges. In *Vicia faba*, maximum growth stimulation occurred with the application of 20 g ae ha^−1^ [19]. In sugarcane and eucalyptus, glyphosate application in the range of 5.8 to 19 g ae ha^−1^ increased the leaf dry mass, stem dry mass, and total dry mass [20]. Glyphosate application at 1.8 and 3.7 g ae ha^−1^ increased the leaf area of sugarcane [21] and eucalyptus [5], respectively. The application of low doses also increased biomass accumulation in maize, soybean, pine [5], and *Brachiaria* [22]. In weeds, Nadeem et al. [2] observed an increase in plant height in the species *Coronopus didymus*, *Chenopodium album*, *Lathyrus aphaca*, and *Rumex dentatus* with the application of 16 g ae ha^−1^. Considering the lowest dose of glyphosate, recommended in the package insert, for weed control (360 g ae ha^−1^), those doses that caused a stimulating effect represent 0.5 to 6% of the full dose.

In the international market, an increase of 4% in world coffee production is estimated for 2022/2023 [23]. Additionally, to guarantee the maximum productive potential of the crop, it is necessary to minimize possible interferences that quantitatively and qualitatively affect the growth and production of the plants [24]. In coffee, glyphosate is widely used for weed management, and because it is not selective, it is applied in a directed jet to avoid contact of the product with the crop [25,26]. Although this technique provides greater application safety, it is common for plants to be exposed to the herbicide, and depending on the rate at which the plants are exposed, symptoms of intoxication may appear, such as chlorosis, narrowing of the leaf blade, growth arrest, and even reduction in the production of grains [25,26,27,28,29], or stimulation of plant growth and physiology may occur [30,31,32].

Thus, the objective of this study was to evaluate the influence of low doses of glyphosate in promoting stimulation in the morphology, physiology, and biochemistry of coffee plants and, with the use of mathematical models, define doses and ranges of doses that characterize the hormetic effect of the herbicide.

## 2. Results

### 2.1. Morphological and Growth Analyses

Coffee plants that presented injuries after glyphosate application developed chlorosis and thinning of the youngest leaves, death of the apical meristem, and growth arrest. The onset of phytotoxicity symptoms was according to the dose to which the plants were exposed, at which higher doses (720, 1440, and 2880 g ae ha^−1^) resulted in death of the apical meristem and growth arrest or decrease, and lower doses (45, 90, 180, and 360 g ae ha^−1^) showed chlorosis and/or leaf thinning. When analyzing the set of symptoms and assigning levels of injuries, phytotoxicity of up to 84% was observed at the 2880 g ae ha^−1^ dose at 42 DAA. The mildest symptoms were observed from the of 33.8 g ae ha^−1^ dose, with 5% injury (Figure 1).

During the growth and development of coffee plants, specifically for the variables leaf number and plant height, the hormetic effect of glyphosate was observed in the evaluations with the longest time since application, with data conforming to Model 1 in the evaluations at 21, 35, and 42 DAA for the number of leaves and 28, 35, and 42 DAA for plant height (Appendix A). Thus, the maximum stimulation rose from 5.6% at 21 DAA to 12.2% at 42 DAA for the number of leaves and from 5.4% at 28 DAA to 10.2% at 42 DAA for plant height (Figure 2). In addition to the stimulation peak, an increase in the dose range was also observed, with an increase in at least 5% of these variables.

For the number of leaves, at 21 DAA, the dose range was 2.8 to 9.8 g ae ha^−1^, with a maximum increase for the 5.7 g ae ha^−1^ dose. At the end of the experiment (42 DAA), the maximum increase occurred at the 29.9 g ae ha^−1^ dose, and the dose range that increased the number of leaves was from 6.7 to 52.8 g ae ha^−1^ (Figure 2). At 28 DAA, an increase in plant height in the range of 7.4 to 20.5 g ae ha^−1^ was observed, with a maximum increase at the 13.1 g ae ha^−1^ dose. At 42 DAA, the maximum height increase was obtained at the 21.5 g ae ha^−1^ dose; however, between the 4.6 and 49.1 g ae ha^−1^ doses, an increase of at least 5% was observed compared to the plants that did not receive herbicide application (Figure 2).

Fitting the data to Model 1 (Appendix A) revealed that the maximum increase in the accumulation of leaf, stem, and total dry mass was 14.8%, 13.6%, and 14.9%, respectively, compared to plants not treated with glyphosate. The doses that promoted these results were 17.0 g ae ha^−1^ for leaf dry mass (LDM), 14.5 g ae ha^−1^ for stem dry mass (SDM), and 17.0 g ae ha^−1^ for total dry mass (TDM). However, for these variables, stimulation of at least 5% was observed at doses up to 47.7 g ae ha^−1^ for LDM, 54.4 g ae ha^−1^ for SDM, 51.0 g ae ha^−1^ for TDM. For leaf area, stimulation occurred up to the 45.3 g ae ha^−1^ dose, with a maximum increase at 19.9 g ae ha^−1^, which represents a 19% increase in leaf area compared to untreated plants (Figure 2).

### 2.2. Physiological Analyses

The results presented in the morphological analyses are consequences of the physiological and biochemical processes involved in the growth and development of plants when exposed to glyphosate. In this sense, the variables inherent to gas exchange and chlorophyll fluorescence largely showed the hormetic effect of glyphosate (Appendix A). Thus, stimulation of the variables CO_2_ assimilation (A), transpiration (E), and stomatal conductance (gs) was observed from the initial moments after herbicide application and lasted until the end of the evaluations (Figure 3, Appendix A).

With the occurrence of the hormetic effect of glyphosate on the CO_2_ assimilation of coffee plants and comparing to the Model 1 data (Appendix A), at least 5% stimulation was observed between the of 0.5 and 71 g ae ha^−1^ doses, with a maximum increase of 24.2% at the 13.6 g ae ha^−1^ dose at 7 DAA. At 21 and 35 DAA, the maximum stimulation was 39.6 and 38.9% at the 14.4 and 21 g ae ha^−1^ doses, respectively. During these periods, there was an increase in assimilation from 0.3 to 128.8 g ae ha^−1^ at 21 DAA and from 0.9 to 71.9 g ae ha^−1^ at 35 DAA (Figure 3).

For transpiration, in the first evaluation (7 DAA), an increase was observed between the 0.3 and 47.3 g ae ha^−1^ doses; however, the 6.3 g ae ha^−1^ dose promoted the maximum transpiration rate, with an increase of 18.9% compared to plants not treated with glyphosate. This same behavior was observed at 21 and 35 DAA, but the stimulation range was between 0.7 and 46.9 g ae ha^−1^ at 21 DAA and between 1.7 and 44.1 g ae ha^−1^ at 35 DAA. The maximum stimulation at 21 DAA (16.3%) was observed at the 9.5 g ae ha^−1^ dose, and at 35 DAA (22.1%) at the 17.8 g ae ha^−1^ dose (Figure 3). Although the increase in transpiration indicates that plants lose more water during the CO_2_ assimilation process, in this case, it is not characterized by plant inefficiency due to the increase in assimilation and the maintenance of water use efficiency.

Glyphosate application provided conditions in which stomatal conductance increased or decreased. At 7 DAA, between the 0.2 and 49.5 g ae ha^−1^ doses, stomatal opening was greater, and the 4.4 g ae ha^−1^ dose promoted the highest conductance values, with an increase of 31.4% compared to plants that did not receive glyphosate application. In the last evaluation, at 35 DAA, an increase in stomatal conductance was observed over the dose range of 2.1 to 44.3 g ae ha^−1^, with maximum stimulation of 17.8% at the 17.5 g ae ha^−1^ dose (Figure 3).

In plants exposed to glyphosate doses that did not negatively affect or promote CO_2_ assimilation, increased CO_2_ accumulation did not occur internally compared to plants without herbicide application, as indicated by the Ci values, which reflect higher carboxylation efficiency. Although the carboxylation efficiency at 7 DAA did not conform to the hormesis model, it was observed that up to the 57 g ae ha^−1^ dose, there were no reductions in the W/Ci ratio greater than 5% (Figure 3). However, at 21 and 35 DAA, the carboxylation efficiency was higher compared to plants not treated with the herbicide by up to 54.5 and 58.7%, respectively, at the 14.4 and 22 g ae ha^−1^ doses, respectively (Figure 3).

The water use efficiency was measured from the amount of carbon assimilated by the plant for each unit of water lost in the processes of transpiration and stomatal opening and closing. In this sense, the effective water use efficiency (EWUE) was calculated by the ratio between CO_2_ assimilation and transpiration, and the intrinsic water use efficiency (IWUE) was defined by CO_2_ assimilation and stomatal conductance. Thus, in both approaches to water use, it was observed that the application of different doses of glyphosate caused different behaviors in coffee plants, but only the intrinsic efficiency conformed to the model that characterizes the herbicide’s hormetic effect. The increased transpiration rate generated by glyphosate application did not cause a decrease in the EWUE, since the increase in CO_2_ assimilation was higher (Figure 3), indicating that a greater amount of CO_2_ was absorbed at the expense of reduced water loss. In this sense, a decrease of more than 5% in the EWUE compared to plants not treated with glyphosate occurred at the 235 and 231.5 g ae ha^−1^ doses at 7 and 21 DAA, respectively. At 35 DAA, there was no significant difference in glyphosate doses, and therefore, none of the models were applied (Appendix A).

With respect to the IWUE, it is understood that the water saving during gas exchange is related to reduced stomatal conductance, but the increase in conductance observed in this study was not reflected in a loss of water use efficiency, since there was an increase in CO_2_ assimilation; thus, at certain doses of glyphosate, the IWUE remained similar or increased compared to plants that were not treated with the herbicide. At 7 DAA, up to the 248 g ae ha^−1^ dose, there was no reduction in IWUE greater than 5%. At 21 and 35 DAA, there was an increase in the IWUE of up to 11 and 24.8% at the 82 and 35 g ae ha^−1^ doses, respectively. However, between the 18 and 160.7 g ae ha^−1^ doses at 21 DAA and between the 1.7 and 170 g ae ha^−1^ doses at 35 DAA, an increase of at least 5% was observed (Figure 3).

The ETR/A ratio, which indicates possible alternative electron pathways, did not conform to the proposed models, but it was observed that coffee plants that were exposed to the lowest doses of glyphosate presented ETR/A values lower or close to those of the untreated plants, regardless of the evaluation period, which indicates greater efficiency in the consumption of electrons in the process of CO_2_ fixation; on the other hand, at higher doses, there was an increase in the ratio, which indicates lower efficiency, since the increase in the rate of electron transport was not proportional to CO_2_ assimilation (Figure 3).

Chlorophyll fluorescence, measured by the ETR and ΦPSII, was affected by glyphosate application. With the exception of the evaluation at 21 DAA, each period conformed to the model characterizing the hormetic effect of the herbicide (Appendix A). In this sense, at 7 DAA, glyphosate application between 3.82 and 197.2 g ae ha^−1^ increased ETR by at least 5%, with a peak of 14.7 at the 50 g ae ha^−1^ dose. This same behavior occurred for ΦPSII, in which the dose range causing stimulation was 5.1 to 194.5 g ae ha^−1^, with a maximum increase of 14.8% at the 55 g ae ha^−1^ dose. At the end of the experiment (42 DAA), the hormetic effect of glyphosate was still observed; however, compared to the initial period, the stimulation range was reduced to the range 2.5 to 55.3 g ae ha^−1^ with a peak at the 16.5 g ae ha^−1^ dose, which reflects a 10.6% increase in ETR and ΦPSII (Figure 4).

### 2.3. Biochemical Analyses

The application of different glyphosate doses in coffee plants generally caused significant responses in the internal concentrations of the herbicide and compounds related to the shikimic acid pathway (Appendix A). The concentrations of glyphosate and shikimic acid in coffee leaves at 7, 21, and 42 DAA did not conform to the proposed models; however, there was significance to the application of different herbicide doses (Appendix A). The highest concentrations of glyphosate in coffee leaves were observed at 7 DAA, where the increase corresponded to the dose applied. During the other evaluation periods, the glyphosate level gradually decreased. At 21 and 42 DAA, no glyphosate levels were observed up to the 90 and 360 g ae ha^−1^ doses, respectively; above these doses, the internal concentration was proportional to the increase in the applied dose (Figure 5). Similar to glyphosate, the increase in shikimic acid levels in coffee leaves was proportional to the dose to which the plants were exposed, even with a decrease in concentration with plant growth.

Analysis of the compounds involved in the shikimic acid pathway after glyphosate application indicated herbicide hormesis on quinic, salicylic, caffeic, and coumaric acids (Appendix A). The hormetic effect of glyphosate on the concentrations of quinic acid in coffee leaves was observed at 7 DAA, with a stimulation of at least 5% in the dose range between 0.1 and 184 g ae ha^−1^ and a maximum stimulation of 58.3% at 4.6 g ae ha^−1^. The concentration of salicylic acid in coffee leaves was higher than in untreated plants at 7 and 42 DAA. Thus, at 7 DAA, the maximum accumulation was 60.5% at the 11 g ae ha^−1^ dose and 28.7% at the 7.2 g ae ha^−1^ dose at 42 DAA (Figure 6).

Although absolute levels of caffeic acid increased after the application of low doses of glyphosate at 7 and 21 DAA, only at 42 DAA did the data conform to the herbicide hormetic effect model (Appendix A). Thus, at 7 DAA, the highest caffeic acid concentration occurred at the 90 g ae ha^−1^ dose, and at the other doses, there was no difference compared to the untreated plants. At 21 DAA, up to the 225 g ae ha^−1^ dose, there was no reduction in caffeic acid in coffee plants by more than 5%. In contrast, at 42 DAA, between the 12.2 and 353 g ae ha^−1^ doses, there was an increase in caffeic acid of at least 5% compared to plants not treated with glyphosate and a maximum accumulation of 27.7% at the 140 g ae ha^−1^ dose (Figure 6).

The application of low doses of glyphosate caused an increase in the levels of coumaric acid in coffee plants at 7 and 21 DAA. Thus, an increase of up to 82.6% was observed with the application of 3 g ae ha^−1^ at 7 DAA and 33.1% at the 53 g ae ha^−1^ dose at 21 DAA. However, between the 0.02 and 251 g ae ha^−1^ doses and from 2.7 to 159 g ae ha^−1^, there was an increase of at least 5% at 7 and 21 DAA, respectively (Figure 6).

For the concentrations of ferulic, benzoic, and chlorogenic acids in coffee plants, only the former was affected by glyphosate application in the three evaluation periods (Appendix A). In comparison with plants that were not exposed to glyphosate, higher concentrations of ferulic acid were observed at 7 DAA for the 11.25 g ae ha^−1^ dose and at 42 DAA for the 1440 and 2880 g ae ha−^1^ doses. At 21 DAA, the highest concentrations were found for the 11.25 g ae ha^−1^ dose (Figure 7).

At 7 DAA, only plants that were exposed to the 11.25 and 180 g and ha^−1^ doses exhibited higher benzoic acid concentrations compared to plants that were not exposed to the herbicide, while at 21 and 42 DAA, benzoic acid levels did not change with glyphosate application (Appendix A; Figure 7). The opposite behavior to benzoic acid occurred for the concentration of chlorogenic acid, in which there was no change in its concentration after herbicide application in the first two evaluations and even at 42 DAA, despite the significant effect of the different glyphosate doses (Appendix A). Compared to the plants that did not receive the product, there was no difference.

Evaluation of the aromatic amino acids revealed that glyphosate application caused changes in the concentrations of phenylalanine, tyrosine, and tryptophan in coffee leaves; however, it was not possible to fit the data to the proposed equations, with the exception of phenylalanine at 42 DAA (Appendix A). With glyphosate application, the production of aromatic amino acids, especially in sensitive plants, was expected to be reduced. However, the concentrations of phenylalanine, tyrosine, and tryptophan were higher at the higher glyphosate doses than in plants that were not exposed to the herbicide (Figure 7).

Although the amino acid phenylalanine is a precursor to salicylic, caffeic, and coumaric acids, the phenylalanine levels differed from those of the other acids, especially at 7 and 21 DAA, since the highest concentrations were observed at higher glyphosate doses, 1440 g ae ha^−1^ at 7 DAA and 2880 g ae ha^−1^ at 21 DAA. At 42 DAA, the concentration of phenylalanine was reduced at the highest doses, and up to the 41.8 g ae ha^−1^ dose, there were no reductions greater than 5% compared to untreated plants (Figure 7). The behaviors of tyrosine and tryptophan were similar to that of phenylalanine, and tryptophan at 42 DAA was not influenced by the glyphosate dose (Appendix A).

In view of the above, the application of glyphosate has a significant influence on the growth and development of coffee plants, and exposure to low doses promotes improvements in the morphology, physiology, and biochemistry of the plant. In this regard, Table 1 presents a summary of the hormetic effect of glyphosate on coffee, where the doses that promoted the greatest stimulation of the morphological variables varied from 14.5 to 30 g ae ha^−1^, depending on the parameter analyzed. In the physiological analyses, the dose variation ranged from 4.4 to 55 g ae ha^−1^, and the biochemical variables ranged from 3 to 140 g ae ha^−1^. In the same sense, the maximum stimulation caused by the exposure of coffee plants to these doses ranged from 10.2 to 26.2% for the morphological variables, from 14.7 to 58.7% for the physiological variables, and from 27.7 to 82.6% for the biochemical variables. The hormetic effect of glyphosate occurred throughout the evaluation period, from 7 to 42 DAA. The duration and consistency of this behavior are reflected in the morphological variables, as the maximum stimulation of all variables occurred in the final period of the experiment, at 42 DAA (Table 1).

## 3. Discussion

The occurrence of glyphosate symptoms in coffee is associated with the concentration to which the plants are exposed. As mentioned above, the symptoms range from chlorosis and thinning of the leaf blade to the death of the apical meristem and reduced or arrested growth. When evaluating simulated glyphosate drift on Arabica coffee, França et al. [25,26] and Schrübbers et al. [29] also observed the occurrence of these symptoms at doses close to those found in this study. In studies by França et al. [25,26], phytotoxicity symptoms emerged from a 56.7 g ae ha^−1^ dose; however, at doses higher than 230 g ae ha^−1^, greater plant growth impairment occurred. Schrübbers et al. [29] observed symptoms starting at 28 g ae ha^−1^.

In addition to not manifesting symptoms of injury, the exposure of coffee plants to lower doses of glyphosate promoted increases in morphological, physiological, and biochemical variables (Figure 2, Figure 3, Figure 4 and Figure 6 and Table 1). This phenomenon, characterized as hormesis, is defined as the stimulatory effect of a subtoxic level of a toxin, which is usually found in a narrow dose range before the onset of the negative effects of the substance [3,5,6,33,34,35,36]. The use of more specific models allows a better understanding of the results generated from more complete dose–response curves, ensuring the detection of hormesis and its implications, such as the maximum stimulation generated, the concentration that produces this effect, and the concentration range that produces a stimulatory response [6,35].

When evaluating the hormetic effect of glyphosate on coffee plants, Carvalho et al. [30] found that this phenomenon is linked to the growth stage in which more developed plants presented stimulation, as variables such as the number of leaves, plant height, and biomass accumulation increased. However, unlike the results obtained in this study, the authors found maximum stimulation at higher doses, 738 g ae ha^−1^ for the number of leaves and 620 g ae ha^−1^ for the plant height.

Although the largest number of studies characterizing the hormetic effect of herbicides on plants are linked to morphological traits, the stimulation of physiological and biochemical variables can also occur at low doses [1]. In this sense, this study observed the stimulation of variables related to gas exchange (CO_2_ assimilation, transpiration, stomatal conductance, carboxylation efficiency, and water use efficiency) and chlorophyll fluorescence (ETR and photochemical efficiency of photosystem II).

Whilst evaluating the influence of glyphosate drift on the physiology of coffee plants, some authors observed an increase in gas exchange variables. Carvalho et al. [31] reported an initial increase in CO_2_ assimilation, transpiration, and stomatal conductance of young coffee plants with applications of 180, 360, and 720 g ae ha^−1^; however, these effects were not persistent, as no increase was observed at 60 DAA. Additionally, for these variables, Domingues Júnior [32] observed stimulation with the application of 7.2, 72, and 360 g ae ha^−1^.

In other plant species, the application of low doses of glyphosate also promoted greater efficiency of variables related to gas exchange. In sugarcane, CO_2_ assimilation, transpiration rate, and stomatal conductance presented maximum stimulation with 6.1, 3.4, and 3.4 g ae ha^−1^ application, respectively [20]. These same authors observed similar behavior for eucalyptus, but the stimulation occurred at doses of 11.6, 11.4, and 11.2 g ae ha^−1^, respectively. In barley, Cedergreen and Olesen [15] observed an increase in the CO_2_ assimilation rate with glyphosate application at 22 g ae ha^−1^.

The increase in transpiration in coffee plants exposed to low doses of glyphosate could indicate that the plants lost a greater amount of water during the CO_2_ assimilation process; however, this increased water loss is not an inefficiency, given the increased assimilation and maintenance of water use efficiency (Figure 3). In the of CO_2_ assimilation process, water loss is inevitable and mainly regulated by the opening and closing of stomata [15].

Plants with the C3 metabolic pathway, such as coffee, tend to concentrate CO_2_ in the leaf mesophyll in greater amounts, especially when they are under some type of stress, to compensate for possible decreases in CO_2_ assimilation and minimize possible damage resulting from such stress [37]. In plants with this metabolism, carbon fixation is initiated by the carboxylation reaction, which in turn is catalyzed by the enzyme ribulose-1,5-bisphosphate carboxylase (rubisco) [37]. In this sense, the amount of CO_2_ that a plant can assimilate as a function of the internal concentration in the leaf (Ci) can determine the carboxylation efficiency. Thus, it was observed that coffee plants exposed to glyphosate doses that did not negatively affect or promote CO_2_ assimilation did not increase CO_2_ accumulation internally (Ci) in addition to increasing carboxylation efficiency, compared to plants without herbicide application (Figure 3). In this regard, Cedergreen and Olesen [15] also observed that the decrease in Ci as a function of stomatal conductance is probably related to a higher rate of CO_2_ fixation in barley plants treated with glyphosate (22 g ae ha^−1^).

Although some authors report that variables arising from the gas exchange process, especially CO_2_ assimilation, provide more consistent information about the indirect effect of glyphosate on the photosynthetic process than the fluorescence variables of chlorophyll [15,38], it was observed in this study that chlorophyll fluorescence, measured by the ETR and ΦPSII, was affected by glyphosate application, either by reduction at high doses or by stimulation at low doses. Silva et al. [39] evaluated the application of low glyphosate doses in conventional soybean and reported a maximum stimulation of 16% in the ETR at the 1.8 g ae ha^−1^ dose, but up to the 32 g ae ha^−1^ dose, the ETR increased; thus, the authors suggest that ETR evaluation can be used as a nondestructive measure to evaluate the hormetic effect of glyphosate.

Although the ETR and ΦPSII variables have similar behavior, the generated data provide different information. ETR represents the process of electron transfer from PSII to PSI, while ΦPSII can be used to determine the activity rates of PSII, in which higher values of ΦPSII indicate that a greater proportion of light is being absorbed by the chlorophylls of the antenna complex of PSII and is converted into photochemical energy [40,41].

High ETR values do not necessarily imply better photosynthetic efficiency, especially under stress conditions, because an increase in the electron transport rate, without greater CO_2_ assimilation efficiency, may indicate greater activity of alternative electron drains, such as photorespiration and the Mehler reaction [42,43]. In this sense, the ETR/A ratio becomes a measure of the relationship between the electrons generated in photosynthetic electron transport (ETR) and the electrons consumed in the CO_2_ assimilation process (A), i.e., it is a variable that indicates the relative yield of electrons from net CO_2_ assimilation [42].

Thus, coffee plants exposed to the lowest doses of glyphosate presented ETR/A values lower than or close to those of the untreated plants, regardless of the evaluation period, which indicates a greater efficiency in electron consumption in the CO_2_ fixation process. To correlate these results with the efficiency of carboxylation, one can assume greater rubisco carboxylase activity, given the increase in the A/Ci ratio at low glyphosate doses (Figure 3). In contrast, at the highest doses, there was an increase in ETR/A in the three evaluation periods (Figure 3), which may imply a greater activity of alternative electron drains; in this case, because coffee is a C3 plant, electrons are more likely be diverted to the photorespiration process. In addition, high ETR/A values may suggest that unconsumed electrons react with oxygen, increasing the risk of oxidative damage [44].

As previously mentioned, with glyphosate application in coffee plants, a hormetic effect on quinic, salicylic, caffeic, and coumaric acids was observed. In the other compounds involved in the shikimic acid pathway, the internal levels of glyphosate did not show a stimulatory effect on the internal concentration of glyphosate and shikimic acid. The increase in shikimic acid levels is a reflection of the inhibition of EPSPS [45,46], and it is suggested that the decrease in concentration with plant growth is the result of a partial recovery of the enzyme level [29]. Schrübbers et al. [47] characterized the similar behavior of glyphosate and shikimic acid as a linear correlation in coffee plants.

Other authors also observed an increase in shikimic acid in coffee plants after glyphosate application, even at low doses. Domingues Júnior [32] observed an increase at 5 and 15 DAA for doses of 7.2, 72, and 360 g ae ha^−1^. When studying the accumulation of shikimic acid in coffee plants under controlled conditions and in the field as an indicator of plant exposure to glyphosate, Schrübbers et al. [29] observed an increase with the application of 82 and 432 g ae ha^−1^ in a controlled environment; however, at the lowest dose, this behavior only occurred at 14 DAA, whereas at the highest dose, it occurred at 14 and 28 DAA. The authors also observed that at 56 DAA, the shikimic acid level was similar to that of the untreated plants. Under field conditions, the increase in shikimic acid occurred only at the 306 g ae ha^−1^ dose.

Glyphosate application also caused an increase in shikimic acid levels in other plant species sensitive to the herbicide, such as sugarcane [20,21,48], eucalyptus [20], wheat [49], soybean [20,39], and corn [5]. Velini et al. [5] reported this increase in corn and soybean plants even with the application of low doses that promoted growth stimulation. In this sense, the authors suggest that this effect may be somehow related to the inhibition of EPSPS, since no occurrence of the hormetic effect on plant growth was observed in soybean resistant to the herbicide, and there was no increase in shikimic acid with glyphosate application in the dose range of 1.8 to 720 g ae ha^−1^.

When analyzing the compounds involved in the shikimic acid pathway that exhibit a hormetic effect with glyphosate application, the accumulation of quinic acid can be considered a carbon source for the biosynthesis of aromatic compounds and is often found in large amounts in plant tissues [50,51,52]. Similar to shikimic acid, the increase in quinic acid after glyphosate application is due to the accumulation of compounds formed prior to the processes catalyzed by EPSPS [53,54]. The accumulation of quinic and shikimic acids in plant tissues, on the one hand, and the reactions involved in quinic acid metabolism, on the other hand, are probably related to the high demand of these alicyclic precursors for phenolic synthesis [55].

In coffee plants exposed to glyphosate at doses of 0, 7.2, 72, and 720 g ae ha^−1^, Domingues Júnior [32] observed the highest concentrations of quinic acid with the application of 7.2 g ae ha^−1^ at 15 DAA. The increase in chemical acid at low doses was also observed in sugarcane [21], in which the authors characterized this increase as the hormetic effect of glyphosate, where the application of 1.8 g ae ha^−1^ at 35 DAA presented the highest concentrations. In maize, an increase in quinic acid was observed after application of 72 and 720 g ae ha^−1^; however, higher concentrations were observed at the highest dose [56].

Another phenolic compound formed in the shikimic acid pathway, which also presented stimulation in coffee plants subjected to low doses, is salicylic acid, a compound that may protect plants from biotic and abiotic stresses [46,57]. Thus, glyphosate application can alter the levels of salicylic acid in plants, especially in species sensitive to the herbicide, since the inactivation of EPSPS can compromise salicylic acid production, and as seen in this study, the increase in salicylic acid levels occurred at lower doses to the detriment of reduction at higher doses (Figure 6).

Silva et al. [39] evaluated the hormetic effect of glyphosate in conventional soybean and observed that the different doses of glyphosate did not have a significant effect on the concentration of salicylic acid; however, they observed an increase in the absolute content of salicylic acid at doses from 1.8 to 720 g ae ha^−1^, with a maximum increase of 170% at a 3.6 g ae ha^−1^ dose, when compared to the untreated plants. In *Digitaria insularis* plants, the resistant genotypes showed higher concentrations of salicylic acid after glyphosate application [58]. Similarly, certain populations of *Conyza sumatrensis* also showed increases in salicylic acid levels compared to the susceptible population [59]. However, even if reduction occurs by inhibiting the shikimic acid pathway, plants may have an alternative pathway for its synthesis; thus, the production of salicylic acid is dependent on the metabolism of the plant and the stressors to which it is subjected. Therefore, this increase cannot be explained exclusively by herbicide application [60].

Among the hydroxycinnamic acids formed in the shikimic acid pathway, caffeic acid together with ferulic and coumaric acids are the most common in coffee plants [32], and with the application of low doses of glyphosate, stimulation of caffeic and coumaric acids was observed (Figure 6). Other authors have also reported the influence of glyphosate application on these compounds. In young leaves of *Abutilon theophrasti*, glyphosate application at concentrations of 1 and 5 mM did not decrease caffeate levels; however, at the lowest concentration, there was an increase up to 12 DAA, which led the authors to conclude that sublethal treatments with glyphosate do not have lasting effects on the levels of caffeic acid and other phenolic acids, such as ferulic acid [53]. In pea plants, glyphosate application via nutrient solution at a concentration of 53 mg ai L^−1^ caused the accumulation of caffeic acid in the roots [52]. In the same study, the coumaric acid content showed little change with glyphosate application, despite the higher concentration at 10 DAA.

As in the present study (Figure 7), other authors also observed an increase or little change in the concentration of aromatic amino acids after exposure of plants to different concentrations of glyphosate [39,56,58,61,62,63,64,65,66]. A possible explanation for the increase in aromatic amino acids after herbicide application would be a possible regulatory mechanism after EPSPS expression that controls the content and changes in metabolic regulations in the biosynthesis of these amino acids and protein hydrolysis [61,65,66].

In this context, Velini et al. [67] note that it is not always possible to conclude that the synthesis of compounds produced far from the site of action of glyphosate (EPSPS) will be blocked or intensely reduced by herbicide application, since possible control systems of the pathway may offset, at least in part, by the lower synthesis of some intermediate compounds. Therefore, the smaller the number of reactions separating a particular compound of interest from the site of action, the greater the likelihood that its concentrations will be reduced as a result of herbicide application [46].

In the context of the hormetic effect of herbicides on plants, the application of low doses of these products trigger mechanisms that improve plant growth and development. In this work, low doses of glyphosate stimulated the growth, physiology and biochemistry of coffee plants. Jalal et al. [68] describe that the main mechanisms of action of glyphosate inducing hormesis in plants are related to stimuli in root growth, biomass accumulation, yield and plant biochemistry. These mechanisms may allow research to advance to identify processes that lead to greater yield and/or stress tolerance, by the direct application of products or in the development of new products that promote the same processes.

## 4. Materials and Methods

The experiment was conducted in duplicate at the Nucleus for Advanced Research in Matology–NUPAM, located at the Faculty of Agronomic Sciences–FCA, of the São Paulo State University “Júlio de Mesquita Filho” Campus de Botucatu–SP. *Coffea arabica* seedlings cv Catuaí Vermelho IAC144 aged 180 days (from seeding to transplantation) were used and subsequently transplanted into 6 L pots filled with a mixture of soil and the substrate Carolina^®^ (sphagnum peat, vermiculite, and carbonized rice husk) in a 3:1 proportion (soil:substrate). The soil used had the following chemical characteristics: pH (CaCl_2_) = 4.2; OM = 17 g dm^−3^; P = 4 mg dm^−3^; K = 0.57 mmol_c_ dm^−3^; Ca = 4.0 mmol_c_ dm^−3^; Mg = 2.0 mmol_c_ dm^−3^; H + Al^+3^ = 28.0 mmol_c_ dm^−3^. In terms of the physical attributes, sand = 210 mg dm^−3^, clay = 692 mg dm^−3^, and silt = 98 mg dm^−3^.

After transplanting, coffee plants were kept under environmental conditions for 45 days before application, arranged in a completely randomized design and subjected to ten doses of glyphosate: 0, 11.25, 22.5, 45, 90, 180, 360, 720, 1440, and 2880 g acid equivalent (ae) ha^−1^. The doses studied represent from 3 to 678% of the lowest dose of glyphosate recommended for weed control (360 g ae ha^−1^). In each dose of glyphosate, a total of 20 plants were applied, 5 of which were intended for non-destructive evaluations (morphological and physiological analyzes) and 15 for the collection of leaf discs in biochemical analyses, with 5 plants for each collection period. Thus, all variables were analyzed with five replications.

As the experiment was conducted in duplicate, in the first experimental repetition, the minimum and maximum temperature was 18 and 28 °C, respectively, and the accumulated precipitation was 425 mm (Appendix A). At the time of application, the plants had an average height of 15.8 cm and an average number of leaves of 14.6. In the second experimental repetition, the average temperature was 19 °C at minimum and 28 °C at maximum and precipitation accumulated was 313 mm (Appendix A), and the plants had an average height of 16.5 cm and a number of leaves of 15. During the conduct of the experiments (from transplanting to final collection), the soil was maintained at field capacity conditions.

Application of the different herbicide doses was performed using an automated sprayer in a closed environment, with a spray bar equipped with four XR 110.02 vs. nozzles spaced 0.5 m apart and arranged at a height of 0.5 m in relation to the experimental units. The working pressure was 2 kgf cm^−2^, with a speed of 3.6 km h^−1^ and spray volume of 200 L ha^−1^ (Appendix A). The product used in the preparation of the mixtures was the isopropylamine salt formulation of glyphosate (Roundup Original^®^–360 g L^−1^).

### 4.1. Morphological and Growth Analyses

After application, the number of leaves and plant height were evaluated at 7, 14, 21, 28, 35, and 42 days after application (DAA), and at 42 DAA, leaf area and leaf, stem, and total dry mass were evaluated, and the level of injury caused by glyphosate was assessed by visual analysis as a percentage from 0 to 100, where 0 represents the absence of symptoms and 100% represents plant death (SBCPD, 1995). The attribution of grades of injury caused by the application of glyphosate was according to the appearance of symptoms (chlorosis, leaf narrowing, reduction and paralysis of growth and death of the apical meristem). In each experimental repetition, the evaluations were made in five plants for each dose of glyphosate.

The plant height was determined from the stem to the apex, and the leaf area was obtained using a leaf area integrator (LI-3100, Li-Cor Inc., Lincoln, NE, USA). For the determination of dry matter, the samples were placed in a forced air oven at 60 °C for 72 h and then weighed on a precision analytical balance (0.0001 g).

### 4.2. Physiological Analyses

The physiological responses were assessed using variables related to gas exchange and chlorophyll fluorescence. In each experimental repetition, measurements were made on five plants for each dose of glyphosate. Using an infrared gas analyzer (IRGA, PP-Systems^®^, Ciras-3, Amesbury, MA, USA), gas exchange was measured between 9:00 a.m. and 11:30 a.m., through net assimilation of CO_2_ (A), transpiration (E), stomatal conductance (g_s_), internal CO_2_ concentration (Ci), carboxylation efficiency (A/Ci), and water use efficiency at 7, 21, and 35 DAA. The carboxylation efficiency was measured as the ratio between assimilation and the internal concentration of CO_2_. Water use was determined as the effective efficiency through the ratio between CO_2_ assimilation and transpiration (EWUE–µmol CO_2_ mmol^−1^ H_2_O) and the intrinsic efficiency of water use obtained by the ratio between assimilation and stomatal conductance (IWUE–µmol CO_2_ mmol^−1^ H_2_O).

Chlorophyll fluorescence was evaluated by means of the electron transport rate (ETR) and effective photochemical efficiency of photosystem II (Φ_PSII_) at 7, 14, 21, 28, 35, and 42 DAA with the aid of a portable fluorometer (Multi-Mode Chlorophyll Fluorometer OS5p–Opti Sciences, Hudson, NH, USA). The readings were performed for each treatment and replicate at five points per plant on fully expanded leaves. Alternative electron pathways were also evaluated using the ETR/A ratio [40,69].

### 4.3. Biochemical Analyses

For the determination of the biochemical components, in each dose of glyphosate, leaf discs of five plants were collected in all fully expanded leaves at 7, 21, and 42 DAA. The samples were placed in an ultrafreezer at −80 °C for the subsequent analysis and determination of the following compounds: internal glyphosate content, shikimic acid, quinic acid, salicylic acid, ferulic acid, caffeic acid, coumaric acid, benzoic acid, chlorogenic acid, and the amino acids phenylalanine, tyrosine, and tryptophan.

The leaf discs were macerated in a mortar with liquid nitrogen, placed in Falcon tubes (15 mL), and subjected to lyophilization (Lyophilizer Alpha 2–4 LD Plus, Martin Christ, DEU) at a temperature of −70 °C for 72 h. After lyophilization, a 100 mg sample was taken, and 10 mL of acidified water (hydrochloric acid) at pH 2.5 was added. Next, the samples were subjected to ultrasound (Elma–Elmasonic P 180 H, DEU) for 30 min in water at 55 °C and centrifuged at 4000 rpm for 10 min at 20 °C. The supernatant was collected (1.5 mL) and filtered through a 35 Millex HV (Millipore) 0.45 μm filter with a 13 mm Durapore membrane and placed in a 9 mm amber vial (Flow Supply) for subsequent quantification of the compounds by chromatography (LC–MS/MS).

The LC–MS/MS system used is composed of a Shimadzu High Performance Liquid Chromatograph (HPLC), model Prominence UFLC, equipped with two LC20AD pumps, SIL-20AC autoinjector, DGU-20A5 degasser, CBM20A controller system (allows fully automated), and CTO-20AC oven (for column temperature control). The 4500 (Applied Biosystems) hybrid triple quadrupole mass spectrometer is attached to the HPLC. The chromatographic analyses were conducted with a C18 column (Phenomenex Gemini 5μ C18RP 110Å) using an injection volume of 20 μL, with 5 mM ammonium acetate in water (phase A) and 5 mM ammonium acetate in methanol (phase B). The flow rate used was 0.8 mL min^−1^, and the gradient mode started with a ratio of 90:10 (water/methanol), increasing to 5:95 at 4 min and returning to the initial condition at 10 min. The run time was 15 min.

### 4.4. Data Analysis

As the experiment was conducted in duplicate, at different times, the data were independently subjected to analysis of variance, and the test of homogeneity of residual variances was applied [70]. Once the homogeneity between the experiments was confirmed, conjoint analysis was performed, in which the experiments were grouped and an experiment with ten replicates was considered for each dose of glyphosate evaluated. Thus, a total of 400 plants were analyzed, 200 in each experimental repetition.

The injury data were fitted to the nonlinear log-logistic regression model proposed by Streibig et al. [71]:(1)$$y=\frac{a}{\left[1+{\left(\frac{x}{b}\right)}^{c}\right]}$$
where y = percentage of injury; x = herbicide dose (g ae ha^−1^); and a, b, and c are parameters of the equation, where a = asymptote between the maximum and minimum points of the variable, b = dose that provides 50% of the asymptote, and c = slope of the curve.

The evaluation of the hormesis effect of glyphosate on coffee plants was determined using five- and four-parameter nonlinear regression models:

Five-parameter model of Brain and Cousens [72]–Model 1:(2)$$y=c+\frac{d-c+\mathrm{fx}}{1+\exp(b\left(\log\left(x\right)-\log\left(e\right)\right))}$$

Streibig four-parameter model [73]–Model 2:(3)$$y=c+\frac{d-c}{1+\exp(b\left(\log\left(x\right)-\log\left(e\right)\right))}$$

These models are parameterized using a unified structure with a coefficient “b” denoting the slope of the dose–response curve, “c” and “d” denoting the lower and upper asymptotes or limits of the response, and parameter “e” being the logarithm of the inflection point. The addition of the coefficient “f” in Brain and Cousens Model 1 is what differs from Streibig Model 2. This parameter is treated as the stimulation rate, attributing the hormesis effect; however, the stimulation effect is only true when this parameter is different from 0 to 5% probability [74,75]. Thus, to conclude that the variables presented a hormesis effect, the data must be fitted to Model 1.

The fit of the data to the five-parameter model is only performed from the analysis of the difference in the mean squared sum of the regression of the two models. Thus, it is possible to test the mean square associated with the inclusion of the coefficient “f”, with one degree of freedom. If the mean square of Model 1 is greater than that of Model 2, the hypothesis of f = 0 is rejected, and consequently, the occurrence of hormesis stimulation is accepted by applying the equation of Model 1. When the mean square value was not significant, the hypothesis that f = 0 was accepted; thus, it was concluded that there was no hormesis effect, and the four-parameter logistic model was used (Model 2) [5]. For data not adjusted to the proposed models, the averages were plotted with the respective confidence intervals.

The confidence interval (CI) was calculated using the t test for all variables using the equation:(4)$$\mathrm{CI}=\frac{\left(t*\mathrm{SD}\right)}{\surd n}$$
where t refers to the tabulated t value (*p* ≤ 0.05), SD is the standard deviation of the data, and n is the number of samples.

## 5. Conclusions

Glyphosate doses greater than 720 g ae ha^−1^ promoted the death of the apical meristem and the growth of coffee plants. Plants exposed to doses between 45 and 360 g ae ha^−1^ showed chlorosis and/or leaf thinning.

With the application of mathematical models, doses of glyphosate were defined that characterized the hormetic effect of the herbicide on the morphology, physiology and biochemistry of coffee plants.

The morphological variables were stimulated by the application of low doses. The 14.5 to 30 g ae ha^−1^ doses promoted the greatest stimulation. In terms of plant physiology, there was an increase in CO_2_ assimilation, transpiration, stomatal conductance, carboxylation efficiency, IWUE, electron transport rate, and PSII photochemical efficiency at doses ranging from 4.4 to 55 g ae ha^−1^ to express the highest stimulation.

In terms of the compounds produced in the shikimic acid pathway, there was a hormetic effect of glyphosate on quinic, salicylic, caffeic, and coumaric acids, with maximum stimulation between the 3 and 140 g ae ha^−1^ doses depending on the evaluated compound and the season of evaluation.

Linked to these results, further studies are still needed to evaluate the production and yield of coffee crops using low doses of glyphosate.

## Figures and Tables

**Figure 1 plants-12-02249-f001:**
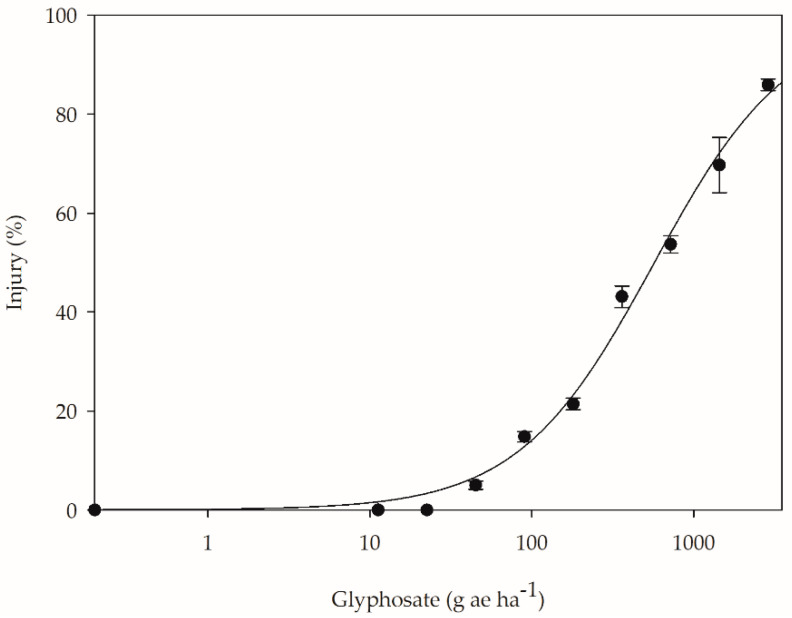
Percentage of coffee plant injury at 42 days after application of different doses of glyphosate.

**Figure 2 plants-12-02249-f002:**
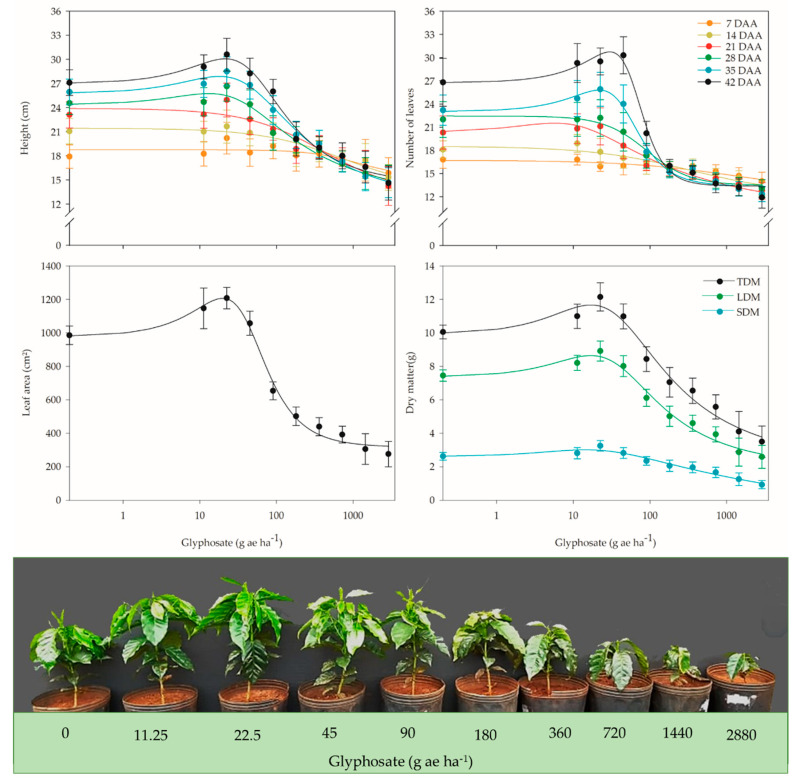
Height (at 7, 14, 21, 28, 35, and 42 DAA), number of leaves (at 7, 14, 21, 28, 35, and 42 DAA), leaf area, total dry mass (TDM), leaf dry mass (LDM), and stem dry mass (SDM) (at 42 DAA) of coffee plants after application of different glyphosate doses. Demonstration of coffee plants at 42 DAA at the studied glyphosate doses. DAA = days after application.

**Figure 3 plants-12-02249-f003:**
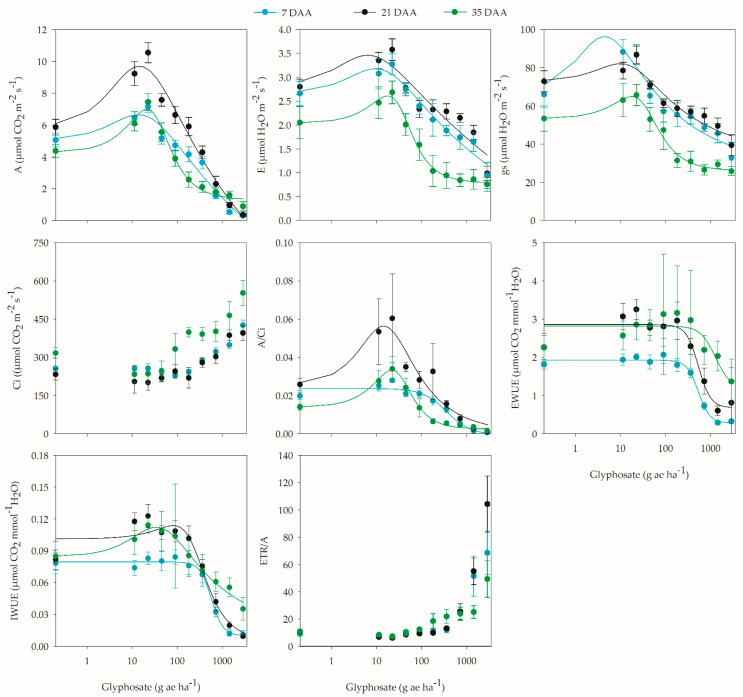
CO_2_ assimilation (A), transpiration (E), stomatal conductance (gs), internal CO_2_ concentration (Ci), carboxylation efficiency (A/Ci), effective and intrinsic water use efficiency (EWUE and IWUE), and ETR/A ratio of coffee plants at 7, 21, and 35 days after application (DAA) of different glyphosate doses.

**Figure 4 plants-12-02249-f004:**
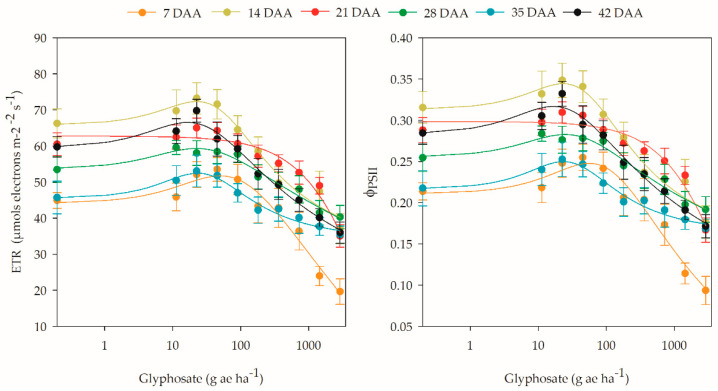
Effective photochemical efficiency of photosystem II (ΦPSII) and electron transport rate (ETR) of coffee plants at 7, 14, 21, 28, 35, and 42 days after application (DAA) of different doses of glyphosate.

**Figure 5 plants-12-02249-f005:**
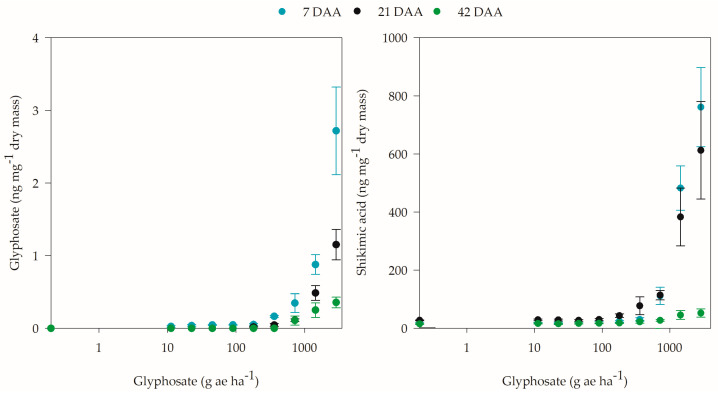
Concentration of glyphosate and shikimic acid in coffee plants at 7, 21, and 42 days after application (DAA) of different glyphosate doses.

**Figure 6 plants-12-02249-f006:**
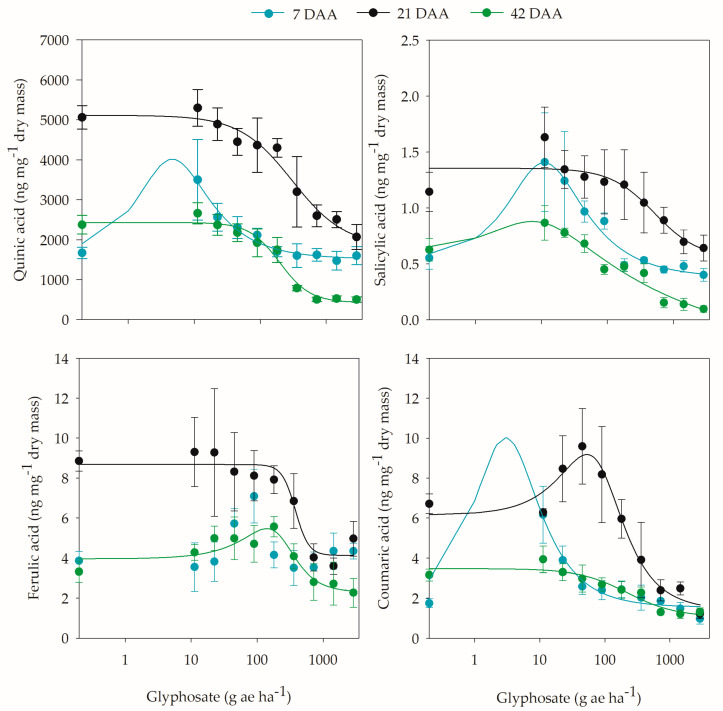
Concentration of quinic, salicylic, caffeic, and coumaric acids in coffee plants at 7, 21, and 42 days after application (DAA) of different doses of glyphosate.

**Figure 7 plants-12-02249-f007:**
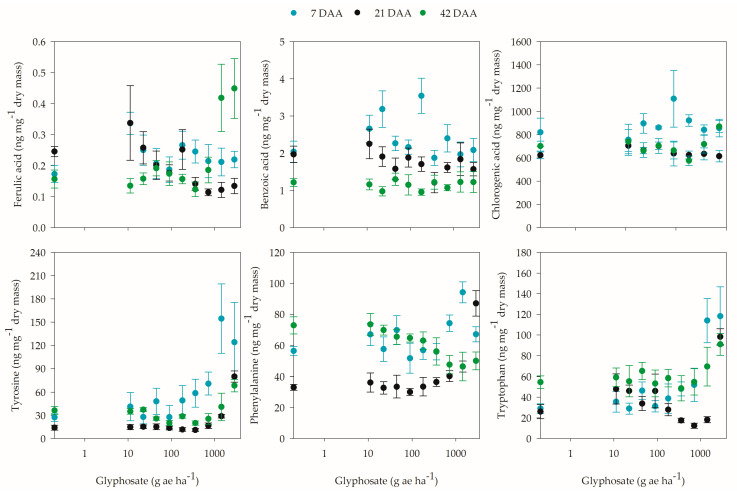
Concentrations of ferulic, benzoic, and chlorogenic acids and of the aromatic amino acids tyrosine, phenylalanine, and tryptophan in coffee plants at 7, 21, and 42 days after application (DAA) of different doses of glyphosate.

**Table 1 plants-12-02249-t001:** Summary of the hormetic effect of glyphosate on the morphology, physiology, and biochemistry of coffee plants.

Variables	Stimulant Dose Range ^1^ (g ae ha^−1^)	Maximum Stimulating Dose ^2^ (g ae ha^−1^)	Maximum Stimulation (%)	Period of Evaluation ^3^
Morphological and growth				
Plant height	4.6–49.1	21.5	10.2	42
Number of leaves	6.7–52.8	29.9	12.2	42
Leaf area	2.3–45.3	19.9	19.0	42
Dry leaf mass	2.0–47.7	17.0	14.8	42
Stem dry mass	1.4–54.4	14.5	13.6	42
Total dry mass	1.9–51.0	17.0	14.9	42
Physiological				
CO_2_ assimilation	0.3–128.8	14.4	39.6	21
Transpiration	1.7–44.1	17.8	22.1	35
Stomatal conductance	0.2–49.5	4.4	31.4	7
Carboxylation efficiency	0.4–87.0	22.0	58.7	35
Intrinsic water use efficiency	1.7–170.0	35.0	24.8	35
Rate of electron transport	3.8–197.2	50.0	14.7	7
Photochemical efficiency of PSII	5.1–194.5	55.0	14.8	7
Biochemical				
Quinic acid	0.1–184.0	4.6	58.3	7
Salicylic acid	0.2–218.0	11.0	60.5	7
Caffeic acid	11.2–353.0	140.0	27.7	42
Coumaric acid	0.02–251.0	3.0	82.6	7

^1^ Dose range that promoted an increase of at least 5% in the analyzed variables compared to plants not treated with glyphosate; ^2^ Dose that promoted the maximum stimulation; ^3^ Period in which the maximum stimulation was observed in days after application (DAA).

## Data Availability

Not applicable.

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
