# Peer review of "Hormetic Effect of Glyphosate on the Morphology, Physiology and Metabolism of Coffee Plants"

_plants, 2023, doi:10.3390/plants12122249_

Round 1

Reviewer 1 Report

plants-2368535-peer-review-v1

 Article: Hormetic effect of glyphosate on the morphology, physiology, and metabolism of coffee plants.

The search idea is good.

Abstracts:

In general, the abstract should be rewritten. For example, the following paragraph has been repeated "Evaluations were performed using the morphological variables height, number of leaves, and biomass accumulation; the physiological variables gas exchange analysis and chlorophyll fluorescence; and biochemical analyses of the internal content of glyphosate and compounds produced in the shikimic acid pathway", in the next paragraph "The hormetic effect of glyphosate on coffee plant morphology was determined by the variables plant height, number of leaves, leaf area, and leaf, stem, and total dry mass".

Introduction:

Line 33-34: "A stimulatory effect caused by low doses of herbicides has been observed in different plant species, whether cultivated or weed[1–8]", A paragraph that weakens the importance of the Manuscript.

The introduction also needs a lot of improvement, as it is very short.

Materials and Methods:

Line 73-76: It is not clear how to calculate the different doses for each pot (plant).

The authors mentioned "spray volume of 200 80 L ha-1", How to apply it to each pot.

Line 82-83 " The experiments were evaluated through morphological, physiological, and biochemical analyses", remove it.

"with one plant per replicate at each dose", the number of pots is not enough for the study.

The authors mentioned "After application, the number of leaves, biochemical components and Chlorophyll fluorescence were evaluated at 7, 14, 21, 28, 35, and 42 days after application", does the effect of the application extend up to 42 days?

Results:

Figure 2: The image of the plants and their age should be separated from the data of plant height, leaf area and dry weight. What is the purpose of merging?

The results are presented in a good way, but I hope the authors try to keep them short.

Discussion:

I also see that the discussion part can be shortened, and it must be clarified whether the coffee plant is susceptible or not to glyphosate.

Conclusions:

The conclusions part needs to be clearly clarified. Do we recommend, for example, spraying glyphosate on coffee plants?

Author Response

We appreciate the contribution. Comments are in the text document and adjustments were made to the manuscript.

Reviewer 2 Report

The article examines the effect of hormesis in coffee plants affected by herbicide glyphosate in different concentrations. High doses of glyphosate caused plant damage, while low ones stimulated growth and improved physiological and biochemical parameters of plants. Mathematical calculations showed the limits of glyphosate hormetic effect on different processes in plants, as well as revealed the glyphosate doses that had the greatest stimulating effect.

It should be noted that this is a well-studied research topic. Thus, in scholar.google.com the query hormetic+effect+of+glyphosate revealed 2290 articles (https://scholar.google.com/scholar?hl=ru&as_sdt=0%2C5&q=hormetic+effect+of+glyphosate&oq=hormetic+effect+), including over 280 articles on coffee plants (scholar.google.com/scholar?hl=ru&as_sdt=0%2C5&q=hormetic+effect+of+glyphosate+AND+coffee+&btnG=).  Therefore, it is obvious that the hormetic effect of glyphosate on plants has long been known and well studied on coffee plants. In this regard, it is unclear what new things the authors of the article were going to say, based on the high degree of study of this topic. If they were trying to prove the limits of hormetic effects, or the mathematical extrapolation of glyphosate effects on different processes in plants, or the varietal response to glyphosate exposure, etc., they should have done it more accurately, starting with the title.

In the Introduction, it was necessary to give a more detailed analysis of the known facts on hormesis in plants affected by herbicides and glyphosate,  and coffee in particular, pointing out the known regularities and existing discrepancies, as well as the need for this study.

 The purpose of the work is presented in a descriptive form, without indicating the hypothesis and the specific tasks implemented in this study.

The section Materials and methods is very poorly written, where there is no information about the age of plants in the experiment, experimental conditions, justification of the choice of glyphosate doses (how much it corresponds to the manufacturer's recommendations), glyphosate concentrations (mg/L) and doses per 1 plant (mg/plant, ml/plant). All these factors, as well as the characteristics of the variety, affect the response of plants to any external influences. The number of plants in the experiment, repetitions (n) for each methods used is not specified in Methods section and Results section. It is not clear how plant damage was determined visually, as a percentage. Is it possible to determine the dry weight of plants after drying at 60 °C?

In the Results section, auxiliary tables 1-6 are given, which are irrelevant in the main text of the paper. They should have been given as Supplementary data. It should be pointed out to the authors that all measurements and calculations given in the article are not absolute, as in plants of different varieties, different age, grown in different conditions - all calculated coefficients will vary greatly, i.e. one can make any number of such calculations and extrapolation curves. Therefore, the scientific component of the presented results is very small.

In the Conclusion, a confirmation is presented that the hormetic effect of glyphosate on coffee plants has been shown earlier, and the effects of glyphosate on the physiological and biochemical characteristics of coffee plants are repeated. However, possible mechanisms of the stimulating effect of glyphosate on physiological and biochemical processes in coffee plants are missing.

The text of the article contains many incomprehensible phrases and incorrect expressions.

The article is not recommended for publication.

Author Response

We appreciate the contribution.
Although he suggested not accepting the manuscript for publication. We make changes based on your suggestions and those of other reviewers. The changes are in the current version of the manuscript.

Reviewer 3 Report

The manuscript entitled "Hormetic effect of glyphosate on the morphology, physiology, and metabolism of coffee plants" intended for publication in Plants is relevant to Special Issue: Plant Biostimulation (Section Crop Physiology and Crop Production). However, I think that manuscript is not suitable for publication in the present form and needs improvements.

Generally, the paper is relatively straightforward and well written,  however, some parts of manuscript require more attention. The Authors should extend the manuscript introduction (among others, add the range of herbicides doses applied to the fields) and modify or more underline the goals of study. The Authors should monitor and provide information on growth conditions of plant material. I suggest to move adequate tables (Tabs. 1-6) from Results to Supplementary material. Some descriptive information from Discussion could be reduce or move to Introduction (e.g. lines: 474-480, 487-496, 523-529, and 618-628); Introduction is rather short now and Discussion is too long. I think also that the Conclusions could be modified. In addition, there are different mistakes in the text of manuscript, including  References list (e.g. lines  22, 70, 71, 82, 110, 209, 212, 262, 307, 336, 364, 461, 474, 495, 496, 526, 539, 558, 578, 588, 691,693, 703, 707, 725, 740, 758, 768, 814, 815, 818, 820) that need to be corrected by Authors. Also the choice of references could be reconsidered and references list must be more carefully checked.

Author Response

We appreciate the contribution. Adjustments were made to the manuscript.

Reviewer 4 Report

The authors studied the effect of glyphosate as hormetic agent to improve the morphology, physiology, and biochemistry of coffee plants. For this purpose, they applied several doses of glyphosate and evaluated numerous parameters including the photosynthetic mechanism and compounds of the shikimic acid pathway. The manuscript provides a sound hypothesis leading to logical scopes and objectives. The Introduction section is quite informative about hormesis and glyphosate action. However, the authors should include information about the importance of coffee and the reasons they selected it for this study, even if the worldwide value of coffee is obviously very high. The experiment is well established and described, and the results are clearly presented. However, I was confused about the glyphosate doses appearing in the results’ text which are not the same as the doses mentioned in the materials and methods. Moreover, I suggest a few amendments about the figure layout. Most importantly, the conclusions are supported by the results and the authors discussed them in a proper manner.

Specific comments are following:

L22. These doses (14.5 and 30 g ae ha-1), as well as in L26 and L28, do not correspond to the doses mentioned in L16 (abstract) and L75 (Materials and Methods). What am I missing?

L23. The values (10.2-26.2% and 14.7-58.7%) are not needed in the abstract section.

L64. Years of duplicate experiments?

L69. You mentioned 3 substrates but only 2 (3:1) parts of the mixture.

L77. Did you spray directly on top of the plants?

L195 (and elsewhere). Similarly to my comment above, 33.8 g ae ha-1 was not mentioned in the tested doses.

L223. In all figures, blue and black lines are not easy to observe. Please change the colours a bit.

L258. The lines are missing in two subfigures (Ci and ETR/A) of Figure 3. The same applies for Figure 5 and 7. Is it on purpose?

Author Response

We appreciate the contribution. Adjustments were made to the manuscript.

Specific comments are following:

L22. These doses (14.5 and 30 g ae ha-1), as well as in L26 and L28, do not correspond to the doses mentioned in L16 (abstract) and L75 (Materials and Methods). What am I missing?

The analyzes were carried out strictly within the adjusted models for each variable and the equations were used to determine the doses that promoted stimuli and characterized the hormetic effect, therefore in the results we bring doses that were not described in the material and methods.

 L23. The values (10.2-26.2% and 14.7-58.7%) are not needed in the abstract section.

Modified in manuscript

L64. Years of duplicate experiments?

The experiment was carried out in two moments, but analyzed jointly. We have added more information in the manuscript.

L69. You mentioned 3 substrates but only 2 (3:1) parts of the mixture.

The ratio of 3 parts of soil to 1 of substrate was used (soil:substrate - 3:1)

 L77. Did you spray directly on top of the plants?

The plants applied in each dose were distributed in the spraying simulator and the application took place over the plants with the application bar at a height of 0.5 m (Figure S1).

L195 (and elsewhere). Similarly to my comment above, 33.8 g ae ha-1 was not mentioned in the tested doses.

L223. In all figures, blue and black lines are not easy to observe. Please change the colours a bit.

Adjusted in the manuscript

L258. The lines are missing in two subfigures (Ci and ETR/A) of Figure 3. The same applies for Figure 5 and 7. Is it on purpose?

For these variables, the data were not adjusted to the models, so we chose not to draw the trend line, leaving only the means of each dose and their respective confidence intervals. We have included this information in the current version of the manuscript.

Round 2

Reviewer 1 Report

Dear authors

Thank you for your deep improvement in your article.  All requested comments were considered.  The article could be published now in the current form.

Author Response

We appreciate the contribution to the improvement of the manuscript

Reviewer 2 Report

The authors have tried to supplement the article in accordance with the comments. The authors accepted most of them, and corrections were made to the manuscript, which significantly improved its understanding. However, some shortcomings remain.

1. I recommend that the authors change the title of the article to include either an estimate of the limits of the hormetic effect of glyphosate, or a mathematical extrapolation of the effects of glyphosate, or a varietal response on coffee plants to glyphosate exposure, etc.

2.  In Section 1. Introduction, paragraphs (2+3) and 4 should be swapped.

3. L. 85-87: 3-phosphoglyceric acid is not an enzyme!

4. L. 118: it is necessary to indicate the age of coffee seedlings at the beginning of the experiment.

5. L. 127 and L. 129-130: ‘with one plant per replicate at each dose’ (l.127), ‘In each dose of glyphosate, a total of 20 plants were applied...’ - eliminate the inaccuracy.

6. L. 143-148: indicate the dose of the solution (mL / plant) and the dose of glyphosate (mg or µg / plant)

7. L. 173: give a reference to the method for determining the dry mass after drying at 60 °C

8. Section 2.4 and Section 3: it is necessary to indicate n (number of plants and replication at determinations) at each experimental point

9. L. 265 and 268: CI and IC - what is right?

10. Discussion: The authors need to suggest possible mechanisms of the stimulating effect of glyphosate on the physiological and biochemical processes in coffee plants

11. Conclusion: L. 896-898 - move to the beginning of the section. For paragraphs (L. 887-895) it is necessary to indicate that these are the approximation of experimental data.

Author Response

We thank you for your contribution to improving the manuscript.

Corrections and comments have been made in the manuscript and in the attached document.

Reviewer 4 Report

The authors sufficiently addressed by comments and suggestions. I suggest publication of the manuscript in Plants. Thank you for giving me the opportunity to assess the manuscript.

Author Response

(The authors gave the same response as above.)

Round 3

Reviewer 2 Report

The authors took into account all the comments on the article and made the necessary corrections or gave a detailed answer to the comments. I think that there are some small inaccuracies in the article, which the academic editor can handle. The article may be accepted for publication.